# Mitochondrial Lipids: From Membrane Organization to Apoptotic Facilitation

**DOI:** 10.3390/ijms23073738

**Published:** 2022-03-29

**Authors:** Aikaterini Poulaki, Stavroula Giannouli

**Affiliations:** 1Hematology Unit, Second Department of Internal Medicine, School of Medicine, National and Kapodistrian University of Athens, 11527 Athens, Greece; aikaterini.poulaki@gmail.com; 2Department of Pathophysiology, School of Medicine, National and Kapodistrian University of Athens, 11527 Athens, Greece

**Keywords:** apoptosis, mitochondria, mitochondrial lipids, cardiolipin, double membrane, cytochrome C, ceramide

## Abstract

Mitochondria are the most complex intracellular organelles, their function combining energy production for survival and apoptosis facilitation for death. Such a multivariate physiology is structurally and functionally reflected upon their membrane configuration and lipid composition. Mitochondrial double membrane lipids, with cardiolipin as the protagonist, show an impressive level of complexity that is mandatory for maintenance of mitochondrial health and protection from apoptosis. Given that lipidomics is an emerging field in cancer research and that mitochondria are the organelles with the most important role in malignant maintenance knowledge of the mitochondrial membrane, lipid physiology in health is mandatory. In this review, we will thus describe the delicate nature of the healthy mitochondrial double membrane and its role in apoptosis. Emphasis will be given on mitochondrial membrane lipids and the changes that they undergo during apoptosis induction and progression.

## 1. Introduction

Cellular biology has traditionally been viewed and therefore researched as a function of the cellular genome and its encoding proteins. However, after the pioneering work of Otto Warburg in the early 1900s [1] and the official dawn of metabolomics much later, in the late 1990s [2], the science of cellular bioenergetics has increasingly become a focus of attention in the scientific community. With its extreme plasticity and potential, metabolism is nowadays the center of pioneering cellular research [3]. To this end, mitochondria function as integrators of bioenergetic status and cell viability. Both classical (including apoptosis and necrosis) and more recently described types of cellular demise (such as ferroptosis and autophagic cell death) are effectively regulated through, and eventually culminate, in mitochondria [4]. Mitochondrial cytochrome C (cytC) release, formation of the apoptosome within the cytoplasm, and cleavage and activation of the caspase cascade effectively mediate both intrinsic and extrinsic apoptotic pathways. While the end result in both pathways—cytC release—is the same, it is mediated by different mechanisms which are still under investigation [5]. Mitochondrial double membrane is the most intriguing feature of the organelle that shall inevitably be disturbed to allow cytC release; it differs from any other cellular membrane both in structure and subsequently in function as well. Moreover, mitochondrial membrane composition changes actively during the apoptotic process and facilitates initiation as well as completion of the phenomenon [4,6].

Although they are a relatively neglected aspect of cellular biology, membranes mediate the vast majority of cells’ functions. Apart from being the barrier that delineates a cell from its surroundings and anchors it to its microenvironment, membranes define different subcellular compartments and maintain diversity within each different organelle, therefore allowing for its unique activities that serve the overall survival and growth [7]. Docking of proteins on membranes facilitates enzymatic reactions as it ensures contact between enzyme and substrate, both of which would otherwise freely float within the cytoplasm. Without the subcellular membranes, each cell would be a mere solution of proteins and metabolites.

More than 50% of cellular membrane mass is composed of lipids [8]. While being essentially the most abundant and impressively variable molecules within the cell, lipids are very poorly studied. Increasing evidence suggests that deregulated lipid metabolism is a common feature of all malignant cells and among others it serves resistance of their mitochondria to apoptotic stimuli [9,10,11]. In an effort to better understand the biology of mitochondrial lipids and their role in mitochondrially induced apoptosis, the intrinsic pathway, we will describe how the mitochondrial lipidome acts in favor of orderly healthy cellular function and survival [11,12]. Thus, after briefly reviewing the exceptional organization of the mitochondrial double membrane, we will dive into its molecular lipid composition. We will discuss how specific lipids that are uniquely featured within the membrane serve the special role of mitochondria as effective energy producers and regulators of the intrinsic apoptotic pathway. We will also present the changes in the membrane’s lipid composition upon initiation of the apoptotic cascade and explain how these changes serve the propagation of the phenomenon. A review of lipids that do not physiologically reside in mitochondria and their role in apoptosis will also be presented.

## 2. Mitochondrial Membrane Physiology and Lipid Composition

### 2.1. The Mitochondrial Double Membrane: Features, Structure and Function

Being the gatekeepers of cellular metabolism, growth and death, mitochondria have a structure that reflects their multitasking routine. Each mitochondrion features two very different membranes—the outer mitochondrial membrane (OMM) and inner mitochondrial membrane (IMM)—that encapsulate two aqueous compartments, the intermembrane space (IMS) and the matrix (Figure 1). Compartmentalization serves a function as specific bioenergetic processes are performed within the matrix, for instance the tricarboxylic acid cycle (TCA) or the beta-oxidation of fatty acids (FAs). Within the matrix also lie mitochondrial DNA and ribosomes where transcription and translation of the few mitochondrially encoded proteins occur [6,13].

The IMS has a much more complex and, as of today, much more elusive role. Protein sorting, redox balancing with glutathione oxidation and reduction, and most importantly, cytC release and apoptotic cascade activation, are some of the many functions regulated by the extremely confined, only 5% of mitochondrial space, IMS [6,13,14]. IMS itself is physically and functionally subdivided into at least two different compartments, the peripheral intermembrane space and the intracristae space (ICS) [15]. The peripheral IMS is a relatively reduced environment with high reduced glutathione (GSH) concentration, where protein and solute trafficking from the cytoplasm to the matrix and vice versa can safely take place [6,16]. Contrarily, ICS is much more oxidized and thus hostile, with yet a crucial function in mitochondrial respiration. The two spaces and the associated IMM are dynamically yet strictly separated through a complex proteinic system anchored either directly or indirectly to the lipids of the IMM (Figure 1) [17]. Protein–lipid and protein–protein interactions ensure that the ICS remains isolated from the rest of the IMS [6]. Disruption of the protein–lipid interactions interferes with the docking of several complexes to the IMM and often occurs during mitochondrial stress that leads to apoptosis.

Mitochondrial cristae are variably shaped invaginations of the IMM within the matrix where the bulk respiratory and ATP synthetic activity of mitochondria materialize [18]. The cristae IMM consists of multiple alternating positive and negative curvature pieces with distinct lipid composition that give the cristae their flask shape appearance [19]. Within this “flask” which constitutes the ICS, reactive oxygen species (ROS) produced during mitochondrial respiration are safely metabolized to water mainly through GSH oxidation (GSSG) [6,18,19]. Of note, in this compartment, the vast majority of membrane unbound (free) cytC floats [6]. The ICS associated-IMM is rich in mitochondrial respiratory complexes [6,20]. The anchoring of specific complexes to confined areas of the IMM allows for optimized electron transport as electron transfer from one complex to another is achieved within a very short distance, thus protecting surrounding molecules from free electrons and the formation of ROS [20]. The association of mitochondrial complexes into higher-order multicomplex structures, the respiratory super-complexes, is achieved through interactions of the proteinic complexes with IMM’s lipids (Figure 1) [20,21]. Distortion of the unique IMM lipidome disturbs this delicate balance and leads to dissociation of super-complexes, reduction in respiratory efficiency, release of electrons, formation of ROS and cellular death [20]. In fact, release of ROS following IMM distortion further deranges the membrane lipidome through ROS-mediated lipid peroxidation (Figure 2), causing additional respiratory perturbations and stimulating a vicious deadly cascade [22]. A more detailed description of the mitochondrial lipidome may as well aid in understanding the exquisite nature of respiration and the procedures that lead from ATP production to ROS release, cytC leakage and apoptosis.

### 2.2. The Lipid Structure of Mitochondrial Membrane: Cardiolipin Abundance, Sphingolipid Paucity and Respiratory Integrity

Based on their molecular structure, membrane lipids can roughly be subdivided into three large categories: phospholipids, sphingolipids and sterols [7]. Phospholipids, which are the most variable group, consist of a glycerol backbone with one—or in the case of cardiolipin, two—phosphate group(s) along with fatty acid (FA) side chains (Figure 3A) [7,24]. Glycerol with the phosphate(s) make up the hydrophilic head of the lipid while the FA tail is hydrophobic with the final structure giving rise to an amphipathic molecule. Attached to the phosphate group, a variety of hydrophilic partners, most commonly choline, serine and ethanolamine, construct heads of different size and charge that subsequently change the biochemical properties and scheme of the whole molecule. Phospholipids are the most abundant membrane lipids [7].

Sphingolipids, on the other hand, consist of a backbone of palmitoylCoA with their hydrophilic head carrying a serine instead of a phosphate partner (Figure 3A) [25]. The palmitoylCoA-serine backbone is the sole sphingoid precursor, called sphingosine. Several acyl chains from FAs will be attached to sphingosine to form sphingolipids including ceramides of variable sizes. Thus, in both phospholipids and sphingolipids, alteration of FA tail length and saturation gives rise to an enormous number of different lipids with different properties. Lipids are exchanged between organelles, transferred to the plasma membrane, and excreted to and/or received by the microenvironment [26]. Moreover, enzymes located within the different organelles actively exchange and modify the FA acyl chains and in the case of phospholipids the phosphate partners, adding another level of variability to local lipidomic regulation [7,27]. Mitochondria, for instance, in close contact with the ER at specific membrane formations called mitochondrial associated membranes (MAMs), receive from the ER phosphatidylserine (PS), which then reform into phosphoethanolamine (PE) that is either incorporated into the mitochondrial double membrane or returned to the ER to give rise to the most abundant membrane phospholipid, phosphatidylcholine (PC) [28,29].

Mitochondria, contrary to most other organelles, show an extreme paucity in both sphingolipids and sterols [8,30]. Their membrane is unique in structure as it consists of two double-layered membranes [6]. Asymmetric distribution of phospholipids serves the specific function of each single membrane with OMM being smooth whereas IMM remains highly folded and compartmentalized (Figure 1) [8]. Being a smooth, lipid-rich envelope, OMM delignates mitochondria from their surroundings and mediates contact with ER. It also serves as the initiating point for mitochondrial kinetic reactions, fission and fusion. The OMM is rich in PC (54% of OMM lipid membrane composition) and PE (29%) [8]. Noticeably, PC is the most abundant phospholipid of all delignating cellular membranes, including the plasma membrane where it consists, along with PE, of almost 80% of its lipid mass [7]. PC’s large hydrophilic head and long acyl chains make the molecule somewhat straight, ideal for bilayer-forming membranes, as two PC molecules can be placed with their heads facing the hydrophilic cytoplasm and IMS accordingly, with no curvature inserted to the bilayer [7].

As opposed to OMM, IMM is rich in protein complexes, including respiratory as well as multiple protein carrying channels and higher-order proteinic structures that achieve the extreme level of compartmentalization of IMM [8,24]. Despite such a high proportion of hydrophilic molecules, IMM has to remain stable and highly impermeable to solutes and cytC in order to maintain the electrochemical potential and ensure efficient respiration and cell survival. Therefore, almost 50% of the IMM lipidome consists of non-bilayer-forming phospholipids, namely PE and the protagonist of its category cardiolipin (CL) [24]. Non-bilayer phospholipids are conical-shaped lipids that confer, due to their conical shape, curvature in the double membrane (Figure 1). Such lipids constitute the lipidomic signature of highly folded IMM (Figure 3B) [7,8,31]. CL is majorly featured in all respiratory membranes including IMM and the evolutionary associated bacterial plasma membrane [24]. It possesses two phosphatidyl moieties that share a glycerol head group. Every phosphatidyl moiety is thereafter connected to another glycerol, and two acyl chains are attached to each side glycerol (Figure 3B). The final, tertiary structure of the molecule is a cone with four hydrophobic tails that are variable in length and saturation and one small hydrophilic head [31].

Because of its conical shape, CL in the IMM monolayer allows for the addition of positive curvature, enabling the folding of IMM and the formation of mitochondrial cristae [24,32]. The IMM parts that are richest in CL are indeed the cristae [6,31]. Alternatively, the curvature can be relieved with the addition of proteins without the membrane integrity being affected. Further stability is achieved through direct interaction of CL with the proteins of IMM, the final result being the highly folded, rich in protein yet stable and leakproof IMM [8]. PE has a very similar shape to CL, and its role as a structural component of the IMM is very much overlapping with CL. Experimental mitochondrial depletion of either PE or CL is not lethal contrary to combined deficiency in both lipids that cause cristae disturbance, dissipation of the electrochemical gradient and cellular apoptosis death [33].

Synthesis of CL takes place within the mitochondria with the PC (and PE) of the OMM and IMM serving as source for the acyl chains [34]. Further modifications to these acyl chains ensure a moderately unsaturated final molecule that gives mature CL its peptide-binding abilities (Figure 4). During the process of CL synthesis and modifications, a CL intermediate is formed with three instead of four acyl chains. This molecule that is called monolyso-CL lacks several of the biochemical properties of mature CL, is less tightly bound to the IMM and, as we will further explain, plays important roles during apoptosis [35,36].

Apart from being a structural component of IMM, CL actively regulates and controls respiratory function, mitochondrial dynamics and cytC release [21,37]. As already mentioned, the efficiency of mitochondrial respiration lies upon the formation of super-complexes [6]. CL directly binds mitochondrial complexes and aids in their higher-order assembly (Figure 1 and Figure 5). Depletion of CL leads to super-complex dissociation that is reversed upon CL repletion [6,38]. Saturation status also plays a major role as the more saturated the CL, the less efficient it is in super-complex assembly [39]. Additionally, CL itself increases the efficiency of the electron transport chain by serving as a proton carrier between the complexes during respiration, a kind of bridge preventing proton release and electrochemical disturbance. Indeed, it is postulated that the presence of CL within the IMM increases the efficiency of electron transport by 35% [24].

CL directly interacts with cytC, docking it in the IMM [6,21]. This interaction is realized by one of two distinct chemical processes, forming two functional pools of cytC within the mitochondrion. Owing to its negatively charged head, CL electrostatically interacts with the positively charged and tertiary folded cytC peptide, thus allowing for a loose cytC-IMM electrostatic “binding” [40]. This loosely bound fraction of cytC performs the electron transfer from complex III to complex IV, a process facilitated by a single electron binding heme moiety within its tertiary structure [41]. The heme element of cytC enables it to act both as an ETC electron carrier as well as a mitochondrial ROS scavenger. When in the highly folded state, cytC’s heme pocket is practically inaccessible to ROS produced during the ETC [42]. Notably, intrusion of one of the four CL’s acyl chain into its hydrophobic pocket confers the hydrophobic binding of cytC hemoprotein to CL [43]. This alternate CL-cytC interplay, mediated by both electrostatic and hydrophobic interactions, firmly confines the protein to IMM [44]. Unable to freely diffuse throughout the membrane, cytC loses its function as an electron carrier for the ETC. In this conformation, the heme pocket of cytC is exposed to, and can therefore interact with, ROS, specifically hydrogen peroxide (H_2_O_2_) produced either de novo or through detoxification of superoxide from superoxide dismutase (SOD) [45]. In either case, IMM confined-cytC participates in redox reactions, binding H_2_O_2_ and acting as a CL peroxidase [46]. Peroxidation of initially one and thereafter more of the four CL’s acyl chains occurs forming oxidized CL (CLox) [40,45]. The highly unsaturated nature of CL further facilitates such interactions [40]. Several ROS scavengers and redox peroxidases act to reduce the CLox back to its steady state [23,46]. This phenomenon allows swift scavenging of ROS and also offers rapid adaptations to changes in redox needs as mitochondria possess a large pool of reduced CL and loosely bound cytC. Due to its decreased affinity for CLox, cytC released from the hydrophobic CL bond reacquires its tertiary structure and joins the IMM loosely bound cytC pool [47].

## 3. Mitochondrial Membrane Lipids in Apoptosis

### 3.1. CL in Apoptosis

Flipping of CL from the IMM to the OMM is a hallmark of upcoming cytC leakage and apoptotic cell death [38,48]. This is because when in the OMM, CL effectively mediates its proapoptotic function primarily by facilitating the contact between cytoplasmic caspase 8 and Bid for the formation of truncated Bid (tBid). tBid on the OMM effectively mediates membrane leakage and cytC release to the cytoplasm [49]. Besides, loss of CL from the IMM confers complete disorganization of mitochondrial super-complexes and release of CL-membrane bound cytC to the IMS from where it leaks into the cytoplasm [38,39,48]. CL also regulates mitochondrial fission through positive interaction with the Dynamin-related protein 1 (Drp1), the major effector of mitochondrial division [37]. Mitochondrial kinetics modulation occurs early in the apoptotic cascade and is essential for its completion (Figure 5) [5,30,37].

While the exact mechanisms that drive CL translocation to the OMM have yet to be described, remodeling of CL’s acyl chains ensues shortly after apoptotic stimuli, with a profound increase in monolyso-cardiolipin (MLCL) (Figure 3B and Figure 5). This CL derivative carries three instead of four acyl chains and normally is quickly metabolized back to CL, for instance through transfer of acyl chains from PC [8]. Inhibition of PC synthesis that happens early in the apoptotic process most likely drives this increase in MLCL [8,36]. MLCL has different properties from CL; is less stable and less efficient in binding proteins; and, due to its structure, may allocate to the OMM (Figure 5). It has been hypothesized that the alterations observed in CL acyl chains after apoptosis induction, such as saturation status, may all be attributed to defects in PC synthesis [36]. On the OMM, CL activates caspase 8 to cleave Bid into tBid. CL also interacts directly with tBid to facilitate its insertion into OMM [50]. Following its docking on the OMM, tBid thereafter activates the proapoptotic proteins Bax and Bak. Their activation enables their oligomerization and organization into OMM channels that allow among others cytC release to the cytoplasm and activation of the caspase cascade (Figure 5) [51]. Of note, it has been shown that Bid in the presence of CL can induce the intrinsic arm of apoptosis by changing the mitochondrial membrane curvature, in the absence of Bax and Bak activation [52]. It is likely that such events are related to the lipid transferase activity of Bid and altered CL IMM-OMM distribution [52,53].

When mitochondrial dysfunction per se drives apoptosis, through overwhelming ROS release for instance, CL undergoes further modifications that may drive and or facilitate cytC release. CytC peroxidase activity does not confine to H_2_O_2_ but also extends to bystander lipid peroxides, using them as substrates for CL peroxidation and thus propagating the process of CLox generation under oxidative stress (Figure 2) [40,54]. When the oxidative stress overwhelms mitochondrial redox mechanisms, the reduction of cardiolipin peroxides back to their native form is inadequate. Susceptibility of CL to oxidation depends on the saturation of its four acyl chains, with the most common form, CL containing four linoleic acid chains—each with two unsaturated bonds—being highly vulnerable to peroxidation [23,40,45]. Notably, CL peroxidation by cytC is well described and uniformly observed early in apoptosis, way before cytC is released into the cytoplasm. This indicates that CL peroxidation is an early step during the intrinsic apoptosis cascade [40,54]. In cytoplasm, cytC interacts with Apaf to cause its oligomerization and initiate the caspase cascade. At the later stages of apoptosis, cytoplasmic cytC also functions as a PS peroxidase. The biochemical changes that follow PS peroxidation lead to the externalization of PS on the plasma membrane where it acts as an “eat me” signal to facilitate effective clearance of the apoptotic cell by the immune system [41]. Additionally, being in close proximity with the ETC and respiratory super-complexes, CL readily serves as a scavenger for electrons that leak from the ETC and can thus become oxidized (CLox) in cytC independent manners [6,49]. Under non-apoptotic circumstances, CLox is converted back to its reduced form by yet ill-described mitochondrial enzymes [34,49]. If electron leakage is more than the mitochondria can compensate for, CLox increases. Having less affinity for cytC, CLox permits its release to the peripheral IMS and eventually to the cytoplasm [34,46] (Figure 1). OxCL can easily detach from IMM and translocate to OMM, where it acts yet again on Bid, promoting membrane leakage [52,55]. Of note, CLox is defective in binding respiratory complexes, leading to further respiratory derangement and being much more efficient in binding caspase 8 and Bid, facilitating the next steps of the intrinsic apoptotic pathway [46].

### 3.2. Sphingolipids and Other Non-Mitochondrial Lipids in Apoptosis

Several non-mitochondrially residing lipids may ignite the intrinsic apoptotic pathway. The best described and those that will therefore be reviewed are the oxidatively truncated phospholipids and a specific sphingolipid, ceramide.

#### 3.2.1. Oxidatively Truncated Phospholipids

Microenvironmental or intracellular conditions can cause aberrant release of ROS. Most of them have to do with overpowering stress on the ETC resulting in the formation of large amounts of ROS, for instance H_2_O_2_ and the most reactive of all ROS, hydroxyl radicals (OH) [56]. OH· thereafter attack the unsaturated bonds of membrane phospholipids (Figure 5). The occurring lipid radical(s) are highly unstable and dissociate into new, truncated lipids coupled with a highly reactive lipid peroxyl radical (Figure 2). The peroxyl radical abstracts a proton (H·) from neighboring membrane phospholipids, establishing an autocatalytic free radical reaction (Figure 5) [22]. Saturated phospholipids can also become oxidized and give rise to truncated forms, with much less efficiency, however [23]. Other ROS and oxidative stressors also have similar effects on phospholipids [23,40,57]. Detailed description of all pathways involved in truncated lipid formation from the several ROS and other redox stressors is beyond the scope of this work.

The occurring truncated phospholipids have a solvent-like effect on the mitochondrial double membrane as they are much more hydrophilic than their precursors. They adduct mitochondrial proteins, causing further respiratory dysfunction and depolarization of mitochondria through mitochondrial outer membrane permeabilization (MOMP) opening [55]. This final event is mediated by the actions of the proapoptotic Bid and can be reversed by overexpression of the prosurvival Bcl2 family member, Bcl-xL [55]. Apoptosis induced by TNFa and Fas ligand involves the action of such truncated phospholipids on mitochondrial membranes [58]. The exact mechanism that leads to the insertion of such phospholipids into the mitochondrial double membrane is yet to be defined [59]. Truncated phospholipids are byproducts of several biological processes, including but not limited to inflammation or mitochondrial dysfunction per se, as well as cellular responses to chemotherapy [60]. If not effectively metabolized, they upregulate the synthesis of ceramides most likely by interfering with the enzymes involved in the process [60]. Their short acyl chains and therefore decreased hydrophobicity enable them to diffuse not only to mitochondria, as already mentioned, but also to the ER where their accumulation causes severe ER stress, which culminates into ER stress—mediated mitochondrial apoptosis [57]. This process may also be effectively mediated by increased ceramide production that does occur and will be analyzed in the following section [61]. It is important to keep in mind that different truncated phospholipids can have added molecule-specific effects on cellular biology and survival [62]. Thus, variability in their molecular structure changes the specificity for several intracellular proteins, distorting their function, and affects the lipids’ hydrophobicity as well as their ability to transfer to different membranes [57]. Despite such an enormous variance, truncated phospholipids uniformly link the extrinsic apoptotic pathway to mitochondrial dysfunction and therefore the intrinsic one [62].

#### 3.2.2. Ceramides

Ceramides are uniformly absent from healthy mitochondria. Their presence in mitochondrial membranes seems to be an event occurring after the induction of the apoptotic cascade [63,64]. Moreover, while their channel-forming ability has been suggested to mediate their destructive force, several experiments have failed to establish ceramide channels on mitochondrial membranes [65]. It has been concluded therefore that their pore-forming ability is highly dependent on the condition employed, thus excluding them as primary apoptotic inducers [66]. However, ceramides do disrupt mitochondrial function and are therefore briefly considered in this review. Depending on the size of their side chains as well as their saturation status, ceramides can either facilitate or protect against apoptotic death [64,67,68]. For instance, short chain ceramides, with C6 being the most prominent example, are considered somewhat water-soluble or at least highly hydrophilic owing to their short hydrophobic chain. They have been shown, much like truncated phospholipids, to increase membrane permeabilization and cause mitochondrial depolarization and therefore death [64]. Longer chain ceramides, on the other hand, such as C18, are extremely hydrophobic. They are abundant in plasma membrane microdomains where they organize into lipid rafts with high rigidity [63]. Nevertheless, when they accumulate in mitochondrial OMM, these properties give rise to their deadly function. Disruption of membrane fluidity disrupts super-complex formations and cristae remodeling, thus causing severe respiratory dysfunction [63,69,70]. Ceramide C18 has also been shown to cause displacement of cytC from the IMM and along with C16 to modulate mitochondrial fission, all events associated with facilitation of the apoptotic cascade [69,71].

Ceramides also activate type 2 protein phosphatase (PP2A), which dephosphorylates several members of the Bcl2 family proteins [72]. Dephosphorylation of Bad and Bax leads to their localization on OMM, increased membrane permeability, opening of MOMP and release of cytC to the cytoplasm [72]. Ceramide-induced activation of cathepsin D also facilitates cleavage of Bid to form tBid that localizes on OMM. PP2A activation also dephosphorylates Bcl2 itself, rendering it inactive [67,72]. Altogether, the changes that the ceramides induce to the Bcl2 family members are in favor of the pro-apoptotic ones and against the prosurvival Bcl2, eventually leading to death. Increases in cellular ceramide contents, along with their sequestration on the ER, cause ER stress [67]. The stress is communicated immediately to mitochondria through numerous MAMs [73]. If persistent, the sustained increase in the intramitochondrial Ca+2 that is released from ER through the MAMs causes mitochondrial swelling and the opening of mitochondrial membrane permeability transition pore (MPTP) [67,74]. MPTP opening is followed by dissipation of the electrochemical gradient, ROS production, cytC release and eventually apoptosis [56,73]. With regards to ceramides, however, one has to keep in mind that the exact effects of the variable ceramides are content-dependent and may be affected by the ratio of ceramide to its precursor sphingomyelin and/or the length and saturation of their acyl chains [64,70,75]. Further referral to the biology and biochemistry of ceramides is beyond the scope of this review.

Other sphingolipids have also been shown to positively affect the apoptotic cascade. Gangliosides GD3 and GM3, for instance, normally reside in cholesterol- and sphingomyelin-rich microdomains at the plasma membrane, but move to the mitochondrial inner membrane in response to extrinsic apoptotic signals. There, they again organize into cholesterol-rich microdomains, disrupting membrane structure and causing collapse of the electrochemical gradient [49]. How and why this transfer occurs remains unknown.

## 4. Conclusions

Lipids are a highly complex group of biomolecules that not only constitute the structural basis of biological membranes but also function as signaling molecules and energy sources. As building blocks of all cellular membranes, they very much determine the membrane’s properties. One of the most complex intracellular membranes with functions in energy production, oxidative stress and cellular death is the mitochondrial double membrane. The mitochondrial membrane lipid composition is unique to serve its multivariate function. Oxidative stress and cellular death are well-described consequences of CL aberrations. In fact, genetic diseases that affect enzymes involved in production and modification–maturation of CL lead to altered mitochondrial structure with defective energy production and premature cellular demise [76]. Disturbed mitochondrial lipid composition is also found in chronic degenerative diseases such as Alzheimer’s and Parkinson’s disease [77]. Moreover, altered lipid metabolism and membrane content is a hallmark of cancer phenotype. Differences in mitochondrial membrane lipid composition and fluidity are being continuously recognized in both solid tumor and leukemia cells. CL with profoundly saturated acyl chains is consistently found in apoptosis-resistant cancerous cell lines [78]. It has been shown recently the preleukemic neoplasms myelodysplastic syndromes (MDS) are characterized by disturbed metabolic profiles with predominance of aberrant lipid metabolism [79]. Manipulation of such differences can render malignant cells more vulnerable to therapeutic approaches, for instance through activation of the apoptotic cascade. Indeed, lipidomics is nowadays a rapidly expanding and very promising aspect of pioneering cancer research. There rises thus the necessity of acknowledging the physiology of mitochondrial lipid composition and its relation to apoptosis in order to better understand pathologic conditions, malignant or not, and to aid in the development of this promising scientific field.

## Figures and Tables

**Figure 1 ijms-23-03738-f001:**
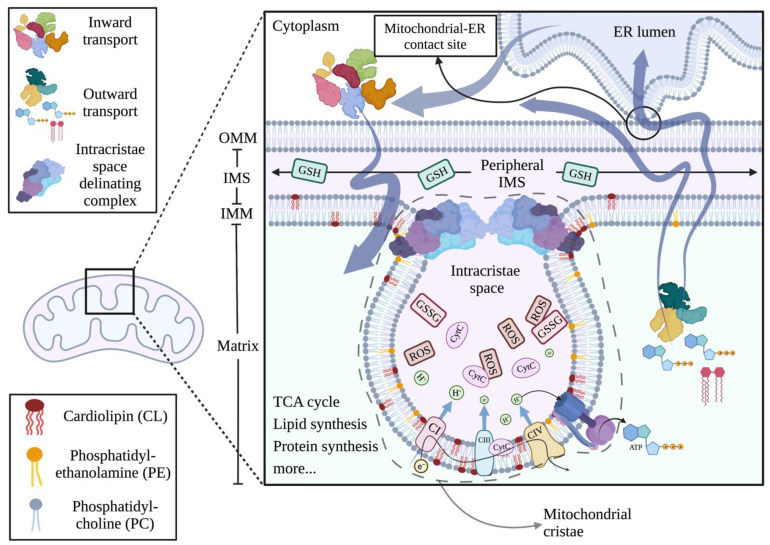
**Mitochondrial double membrane organization.** Schematic presentation of the organization of mitochondrial double membrane with emphasis on membrane lipid composition. OMM, outer mitochondrial membrane; IMS, intermembrane space; IMM, inner mitochondrial membrane; GSH, reduced glutathione; GSSG, oxidized glutathione; CytC, cytochrome C; ROS, reactive oxygen species; CI,III,IV, respiratory complexes 1,3,4. Created with BioRender.com, accessed on 20 March 2022.

**Figure 2 ijms-23-03738-f002:**
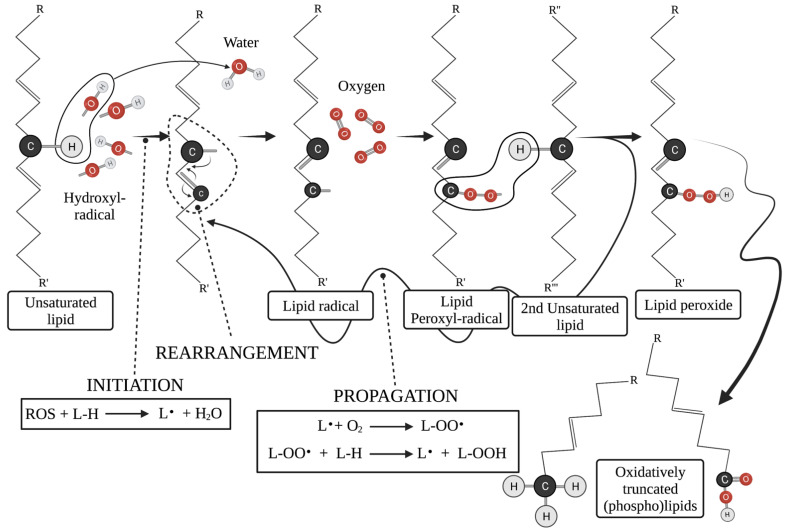
**Schematic presentation of the lipid peroxidation reaction. INITIATION:** During this first step of lipid peroxidation, an initiator being for instance an ROS (in this paradigm a hydroxyl radical (OH^-^)) attacks an unsaturated bond of a Lipid (L) acyl chain. From this reaction, a lipid radical is formed along with water (H_2_O). **REARRANGEMENT:** During this step, a random reorganization of electrons occurs, giving rise to a new lipid radical (L^.^). **NOTE:** This step may or may not occur. **PROPAGATION**: During this step of lipid peroxidation, the lipid radical, rearranged or not, reacts with molecular oxygen (O_2_), forming a lipid peroxyl radical (L-OO^.^). The peroxyl radical then abstracts a proton from a bystander lipid, forming a novel lipid radical that reenters the reaction and a lipid peroxide (L-OOH). The lipid peroxide may thereafter be remodeled into oxidatively truncated phospholipids [23]. Due to their different size, oxidatively truncated lipids have different properties than their precursors. Lipid peroxidation and oxidative truncation also happens on the acyl chains of membrane phospholipids. The subsequent altered scheme distorts membrane integrity with variable effects for the cell. In the case of mitochondria, such consequences are analyzed in detail. C: Carbon, H: hydrogen, O: oxygen (atomic), R-R′-R″-R‴: acyl chain side groups. Created with BioRender.com, accessed on 20 March 2022.

**Figure 3 ijms-23-03738-f003:**
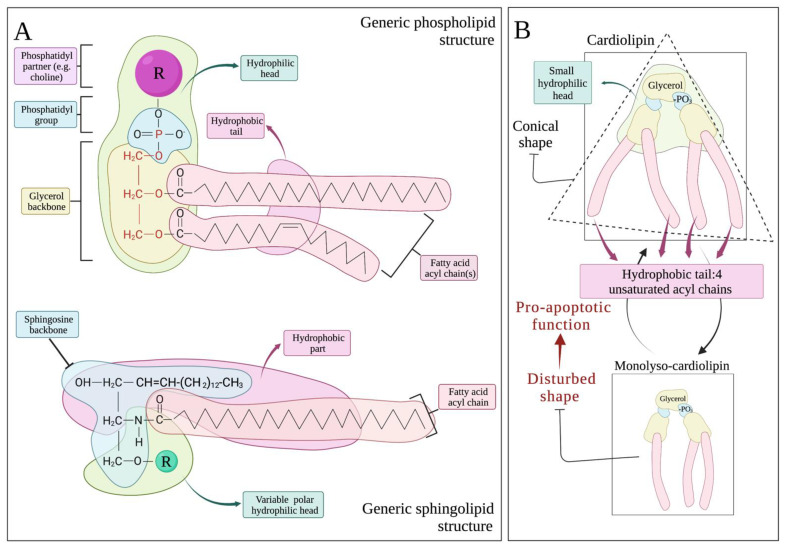
(**A**) **Schematic presentation of phospholipids and sphingolipids generic structure**. (**B**) **Schematic presentation of the conical tertiary structure and rough composition of cardiolipin (CL)**. Monolyso-CL is also shown. Created with BioRender.com, accessed on 21 March 2022.

**Figure 4 ijms-23-03738-f004:**
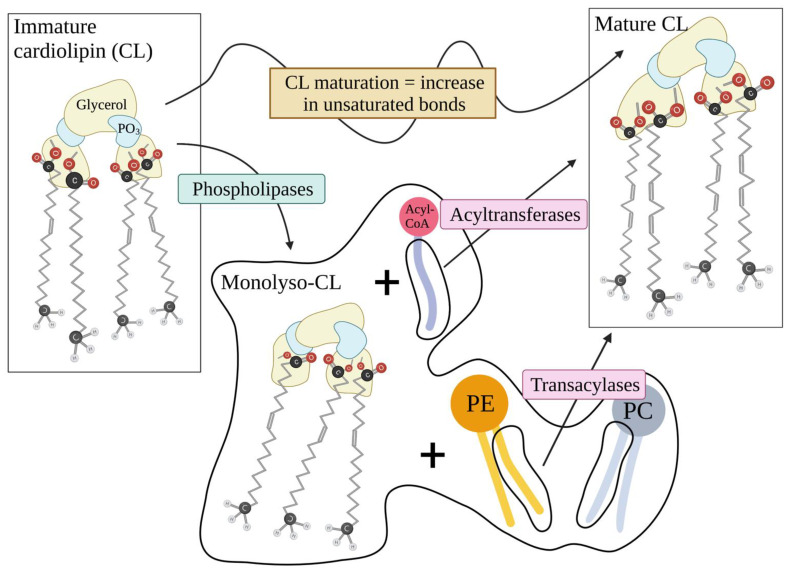
**Schematic presentation of cardiolipin (CL) maturation process.** The maturation process involves the actions of several mitochondrial enzymes that use mitochondrial and endoplasmic reticulum (ER) phospholipids such as phosphatidylethanolamine (PE), or more commonly phosphatidylcholine (PC), and even AcetylCoA to increase the unsaturated bonds of CL’s acyl chains. To achieve this, the immature CL undergoes removal of an acyl chain and substitution if the missing chain by another more unsaturated chain from PE, PC or others. Created with BioRender.com, accessed on 23 February 2022.

**Figure 5 ijms-23-03738-f005:**
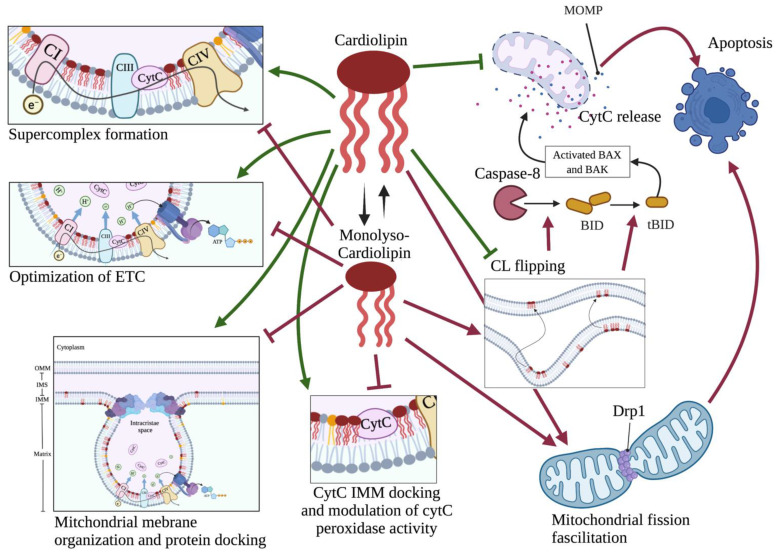
**Effects of cardiolipin (CL) on physiology and apoptosis.** Emphasis is given on CL’s unique ability to directly bind and anchor proteins and CytC on IMM. Formation of monolyso-CL leads to loss of all physiologic abilities of mature CL, flipping to the OMM from the IMM where CL is normally confined and facilitation of apoptosis in a Bid-mediated manner. Of note, loss of mature CL from the IMM disturbs electron transport, leading to increased ROS formation and ignition of the apoptotic cascade. Besides, mature CL loss allows for the membrane-bound CytC to be released to the IMS and sensitizes to membrane permeability and the cell to apoptosis. Created with BioRender.com, accessed on 20 March 2022.

## Data Availability

Not applicable.

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
