# Peer review of "Mitochondrial Lipids: From Membrane Organization to Apoptotic Facilitation"

_ijms, 2022, doi:10.3390/ijms23073738_

Round 1

Reviewer 1 Report

The authors of the paper entitled: "Apoptosis: Focusing on mitochondrial lipids" described the major lipids found in mitochondria. They focus on the substructures of mitochondria, the double membranes and the compartments as subunits of the respiratory chain. The importance, function, and properties of cardiolipin, sphingolipids, and phospholipids were described. Special attention has been given to the description of their role in apoptosis.

This work presents a basic and clear account of the interplay of mitochondrial lipids in the outer and inner mitochondrial membranes.

The figures are descriptive, but in some of them the labelling within the image is too small to be readable, e.g., Figure3. Also, the "disturbed shape" should be corrected in the image.

Author Response

Dear reviewer, 

Kindly find attached our point by point response,

Kindest regards, 

The authors.

Reviewer 2 Report

Poulaki and Giannouli tried to describe in this review the role of lipids in mitochondrial function and apoptosis. A good description of mitochondrial spaces and lipid structures is done, however the role of mitochondrial lipids in apoptosis is lacking and not fully analyzed. For this reason, the manuscript needs major revisions. Major points are listed below.

  1. Indicate the peripheral intermembrane space in the fig 1.
  2. Fig 2 is chaotic and should be better explained
  3. The oxidation of cardiolipin is not fully explored. Other mechanisms can be activated.
  4. At the end of chapter 2.4.1 rhe description of truncated phospholipids and how they induce cell death is superficial and not accurate.
  5. Fig 5 represents the activation of Bax and Bak after tBID and cardiolipin flipping. This is not cited in the text.
  6. In general, the apoptotic signaling induced by mitochondrial lipids should be better described, as it should be the major topic of the review.

Author Response

(The authors gave the same response as above.)

Round 2

Reviewer 2 Report

The authors properly provided responses to the reviewer's questions. The manuscript is now ready for pubblication.